# Crossover of Rate-Limiting Process in Plasma Gel Growth by Contact with Source of Gelator

**DOI:** 10.3390/gels7010011

**Published:** 2021-01-28

**Authors:** Akitsugu Kawabata, Takao Yamamoto, Hiroki Shinoda, Kazuto Yoshiba, Yoshiharu Toyama, Susumu Tanaka, Toshiaki Dobashi

**Affiliations:** 1Division of Molecular Science, Graduate School of Science and Technology, Gunma University, Kiryu 376-8515, Gunma, Japan; t14301051@gunma-u.ac.jp (A.K.); t14301081@gunma-u.ac.jp (H.S.); yoshiba@gunma-u.ac.jp (K.Y.); 2Division of Pure and Applied Science, Graduate School of Science and Technology, Gunma University, Kiryu 376-8515, Gunma, Japan; tyam@gunma-u.ac.jp; 3Faculty of Agriculture, Takasaki University of Health and Welfare, Takasaki 370-0033, Gunma, Japan; toyama@takasaki-u.ac.jp; 4Faculty of Health and Welfare, Takasaki University of Health and Welfare, Takasaki 370-0033, Gunma, Japan; tanaka@takasaki-u.ac.jp

**Keywords:** plasma, gel growth dynamics, contact with source of gelator, rate-limiting process

## Abstract

Plasma is regarded as a solution of precursor polymers specifically transformed to gel-forming polymers by a reaction with initiators. We developed a theory for the gel growth dynamics of plasma induced by contact with a source of gelators that are yielded by the initiation. In developing the theory, we combined the Ginzburg–Landau type dynamics with the gelator diffusion dynamics expressed by the moving boundary picture. The theory predicts the crossover of the rate-limiting process in the time course of the thickness of the gel layer *X* from the energy-limited process expressed by X∼t to the diffusion-limited process expressed by $X∼\sqrt{t}$, where *t* is the time elapsed from when the plasma comes into contact with the source of gelators. A demonstration experiment was performed by placing a tissue factor coating plate as the initiator in plasma. Log–log plot of *X* vs. *t* showed a crossover as predicted by the theory, and the parameters characterizing plasma were determined.

## 1. Introduction

The gelation dynamics triggered by contact of gel-forming polymers with gelators is expressed by scaled equations based on the moving boundary picture for both diffusion-and energy-limited processes [1]. However, the dynamics of gelation for systems consisting of multiple diffusion- and energy-limited processes is complex and has not been clarified yet. One of the most important examples of such system is blood coagulation (gelation), which occurs in the blood by contact with a blood coagulation factor (initiator). The blood consists of corpuscles and plasma, a solution of many kinds of proteins. Among the proteins, most of those that play important roles in blood gelation are polymers called zymogen, which is an inactive precursor of an enzyme (protease) and activated by the corresponding active enzyme specifically transformed from other zymogen. The active enzymes are immediately inactivated by serpins, a superfamily of proteins that irreversibly inhibit the reaction of their target protease by undergoing a large conformational change to disrupt its active site [2]. Other proteins such as protein C are also concerned with the regulation of gelation. Therefore, blood gelation is a complex phenomenon resulting from a cascade of zymogen activation events [3] regulated by serpins and other related proteins. The pathway of blood gelation is traditionally classified into the extrinsic pathway activated by contact with tissue factors and the intrinsic pathway triggered by collagen or polyanions through the activation of the factor IX. Recently, the latter pathway without triggering polyanions was shown to be also activated by microparticles yielded due to hemolysis during blood transfusion [4,5] and erythrocytes in the case of blood flow stagnation, such as in deep vein thrombosis [6,7,8,9]. Both pathways converge into the zymogen activations from prothrombin to thrombin by activated factor *X_a_* and then from fibrinogen to fibrin by thrombin. Finally, fibrin molecules are integrated to form protofibrils and then a three-dimensional network in plasma (plasma gel). Plasma contains about 0.25 wt% fibrinogen, so the gel is assumed to be loose [10] to allow proteins to diffuse in the gel. Though many reaction process and diffusion process proceed serially or in parallel, the blood gelation process can be simplified as follows: the initiation reaction, transformation of fibrinogens to fibrins or protofibrils resulting from the cascade reaction consisting of many reaction-diffusion processes and fibrin network formation in plasma. 

Blood coagulation function tests extract information about blood properties using simplified systems that reproduce the main pathways of the blood coagulation reaction cascade. The clot (gel) growth test using plasma/tissue factor interface has become a standardized procedure using a testing kit called “thrombodynamics” [11]. A number of studies have demonstrated the relationship between some of the parameters characterizing the gel growth behavior and various diseases or changes in blood properties [12,13,14]. In this test, the time dependence of the spatial distribution of thrombin, a key protein to transform fibrinogens to fibrins, is also observed by adding a chemical (7-amino-4-methylcoumarin (AMC) derivative) to plasma where the chemical fluoresces when it is specifically cleaved by thrombin. Tissue factors induce a localized thrombin activity impulse, which propagates in space and possesses all the characteristic traits of a travelling wave. The gel front line travels faster than the thrombin peak and slower than the thrombin front line [11]. This is reasonable as a thrombin concentration above a certain threshold is required for gel formation. The gel growth behavior largely depends on the amount and species of ions and supplemented proteins, such as thrombomodulin; gel growth seems to be linearly proportional or root squared or in a complex manner with time [11,15]. To understand the different behaviors by a theoretical consideration with minimum number of parameters is challenging from a physical aspect for blood coagulation dynamics. 

In this study, we tried to find the mechanism of different gel growth behaviors by considering a simplified model of gel growth from plasma (or precursor solution) that is consistent with the above observed results. Our model deals with a steady process after the initiation reaction yields “a source of gelators” at the contact of plasma with the initiator. Here, it is worth noting that “gelator” is an activated protein (or proteins) that is activated from zymogens as a result of contact of plasma with an initiator such as the tissue factor, and is inactivated after it activates the next step in the reaction cascade to finally convert fibrinogen to fibrin in plasma, while diffusing without consumption in the gel or the fibrin solution. The model consists of two processes; in the first process, the gelators inflow into plasma to activate it, while the second process is the gelation of the activated plasma. We performed a demonstration experiment to analyze the results using theoretical equations. 

## 2. A Theory of the Crossover of the Rate-Limiting Process in Plasma Gelation

As shown in Figure 1, plasma is contained in the rectangular region 0≤x≤L. A solution of precursor polymers (nonactivated plasma) is transformed to a solution of gel-forming polymers (activated plasma) by an influx of coagulation factors (gelators) and becomes gel. The main polymer in the nonactivated and activated plasma is fibrinogen and fibrin, respectively. It is reasonable to assume the front line of the gel-forming polymer solution as well as that of the gel layer as the propagation of thrombin (the final product in the reaction cascade) that transforms precursor polymers to gel-forming polymers can be defined from the spatial distribution of thrombin in observation.

### 2.1. Model of Plasma

To model the plasma, let us introduce the order parameter ϕ expressing the degree of gelation; the plasma is in the sol state when ϕ=0 and in the gel state when ϕ>0. According to the Landau theory for phase transition [16], we describe the sol–gel transition of plasma. In terms of the order parameter ϕ, the local free energy of plasma is expressed as
(1)$$f\left(\phi ;a\right)=g\phi^{2}{\left(\phi -1\right)}^{2}+a\phi^{2}$$
where g is a positive constant, and a is a parameter whose value is changed by an inflow of gelators.

When gelators are absent, the plasma is a precursor solution (nonactivated plasma). The parameter a is chosen as
(2)$$a=a_{s}>0$$
with $a_{s}$ being assumed to be large enough. In this case, the relationship between the free energy and the order parameter is illustrated in Figure 2a. The value of the order parameter minimizing the free energy is ϕ=0. Hence, the plasma is in the sol state. 

When gelators flow in the plasma, the plasma becomes a gel-forming polymer solution. The plasma is activated. The activation is expressed by the value of the parameter a, which is given by
(3)$$a=a_{G}<0$$
where the condition
(4)$$a_{G}>-g$$
is imposed. The free energy is illustrated in Figure 2b. The minima of f exist at ϕ=0 and $\phi =\phi_{+}>0$, and the maximum at ϕ=ϕ_, as shown in Figure 2b. The two values $\phi_{\pm }$ are given by
(5)$$\phi_{\pm }=\frac{3\pm \sqrt{1-\frac{8a_{G}}{g}}}{4}$$

The values of the minima are $f\left(0\right)=0$ and
(6)$$f\left(\phi_{+};a_{G}\right)=\frac{1}{3}g\phi_{+}^{3}\left(2\phi_{-}-\phi_{+}\right)$$

The condition that the gel state at $\phi =\phi_{+}$ is stable is
(7)$$2\phi_{-}-\phi_{+}=\frac{3}{4}\left(1-\sqrt{1-\frac{8a_{G}}{g}}\right)<0$$

This condition is always satisfied by $a_{G}$. On the other hand, the state ϕ=0 is metastable.

As the concentration of gelators is inhomogeneous, the gradient term dϕ/dx should be taken into account and the free-energy functional expressing the whole plasma is given by
(8)$$F=\int_{0}^{L}[\frac{1}{2}\kappa {\left(\frac{d\phi }{dx}\right)}^{2}+f\left(\phi ;a\right)]dx$$
where κ is a small positive constant relating to the free-energy increment due to the inhomogeneity. The time development of the order parameter ϕ is described in terms of the Ginzburg–Landau (GL) equation on the basis of the free-energy functional. As the order parameter is nonconservative, the GL equation is given by [17,18]
(9)$$\frac{\partial \phi }{\partial t}=-\Gamma \frac{\delta F}{\delta \phi }$$
where Γ is the kinetic coefficient for ϕ. Note that because the parameter a depends on the inflow of gelators, a is a function of the position x and the time t, $a=a\left(x,t\right)$. 

When the plasma is activated (the value of a changes from $a_{s}$ to $a_{G}$), the gelation of the plasma starts and the value of the order parameter ϕ changes from ϕ=0 to $\phi =\phi_{+}$ obeying the GL Equation (9). 

### 2.2. Inflow of Gelator

First, let us discuss the first process, in which the inflowing gelators change the parameter a from $a_{S}$ to $a_{G}$. The process can be simply described by the moving boundary picture [1,19]. We assume the following simple assumptions:(1)All the inflow gelators arriving at the precursor solution instantly make precursors into gel-forming polymers and then are inactivated.(2)Gelators are not consumed but diffused in the gel-forming polymer solution.

From assumption (1), in plasma, we have $a=a_{S}>0$ before the inflow of gelators and $a=a_{G}<0$ just after the inflow of gelators. Gel grows immediately after the condition $a=a_{G}$ is satisfied by the GL Equation (9). The flux of inflow gelators in plasma is expressed in terms of the concentration ρ and the velocity $\vec{v}$ as
(10)$$\vec{J}=\rho \vec{v}$$
(11)$$\vec{v}=-k\frac{\partial \mu }{\partial x}\vec{e_{x}}$$
where $\vec{e_{x}}$ is the unit vector along the *x*-axis, k the mobility, and μ the chemical potential of the gelator. As the inflow of gelators can be considered to be in a steady flow, we have
(12)$$div\;\vec{J}=0$$
Expressing
(13)$$\vec{J}=j\vec{e_{x}}$$
with the magnitude of the flux
(14)$$j=-k\rho \frac{\partial \mu }{\partial x},$$
we rewrite Equation (12) as
(15)$$\frac{\partial }{\partial x}\left(-k\rho \frac{\partial \mu }{\partial x}\right)=0$$

Integrating both sides of Equation (15) with respect to x, we have
(16)$$-k\rho \frac{\partial \mu }{\partial x}=C$$
where *C* is the integration constant that does not depend on *x*. The position of the forefront of the gel-forming polymer solution is denoted as $x=x_{f}$, as shown in Figure 1. Integrating both sides of Equation (16) from x=0 to $x=x_{f}$, we obtain
(17)$$-k\int_{0}^{x_{f}}\rho \frac{\partial \mu }{\partial x}dx=Cx_{f}$$

Introducing a function $f_{c}$ of gelator concentration, which satisfies $\mu =\frac{\partial f_{c}}{\partial \rho }$, we derive
(18)$$\int_{0}^{x_{f}}\rho \frac{\partial \mu }{\partial x}dx={\left[\rho \mu -f_{c}\left(\rho \right)\right]}_{0}^{x_{f}}=\pi \left(x_{f}\right)-\pi \left(0\right)$$
where
(19)$$\pi \left(x\right)=\rho \left(x\right)\mu \left(\rho \left(x\right)\right)-f_{c}\left(\rho \left(x\right)\right)$$

Equation (16) reduces to
(20)$$-k\left(\pi \left(x_{f}\right)-\pi \left(0\right)\right)=Cx_{f}$$

From Equation (20), the integration constant *C* is obtained as
(21)$$C=k\frac{\pi \left(0\right)-\pi \left(x_{f}\right)}{x_{f}}$$
and the magnitude of the flux is given by
(22)$$j=k\frac{\pi \left(0\right)-\pi \left(x_{f}\right)}{x_{f}}$$

In Equation (22), $\pi \left(x_{f}\right)$ is determined by the equilibrium of gelators at the front of the gel-forming polymer solution $x_{f}$ so that it does not depend on $x_{f}$. Therefore,
(23)$$\Delta \pi =\pi \left(0\right)-\pi \left(x_{f}\right)=\mathrm{constant}$$
(24)$$j=\frac{k\Delta \pi }{x_{f}}$$
the value Δπ indicates the magnitude of the gelator source.

If we denote the number of gelators required to convert the precursor into the gel-forming polymer per unit volume as $\rho_{G}$, we have
(25)$$jdt=\rho_{G}dx_{f}$$

Substitution of Equation (24) into Equation (25) gives $\frac{k\Delta \pi }{x_{f}}dt=\rho_{G}dx_{f}$. The time development of the front $x_{f}$ is obtained as
(26)$$\frac{dx_{f}}{dt}=\frac{k\Delta }{\rho_{G}}\;\frac{1}{x_{f}}$$

Let t=0 be the time when the plasma made contact with gelators. From Equation (26), we have
(27)$$\int_{0}^{t}x_{f}\frac{dx_{f}}{dt}dt=Kt$$

Integrating both sides of Equation (27), we have
(28)$$\frac{1}{2}x_{f}^{2}=Kt$$
where
(29)$$K=\frac{k\Delta \pi }{\rho_{G}}$$
is a constant. The front of the gel-forming polymer solution moves in proportion to squared time as
(30)$$x_{f}\left(t\right)=\sqrt{2Kt}\;\propto \;\sqrt{t}$$

### 2.3. Gelation Dynamics

Next, let us analyze the second process (the gelation of the activated plasma), in which ϕ changes from ϕ=0 to $\phi =\phi_{+}$. The change (gelation of plasma) is allowed only in the region where the condition $a=a_{G}$ is satisfied, i.e., $x\leq x_{f}$. As the functional derivative in Equation (9) is obtained as
(31)$$\frac{\delta F}{\delta \phi }=-\kappa \frac{\partial^{2}\phi }{\partial x^{2}}+\frac{\partial f\left(\phi ;a_{G}\right)}{\partial \phi }=-\kappa \frac{\partial^{2}\phi }{\partial x^{2}}+4g\phi \left(\phi -\phi_{-}\right)\left(\phi -\phi_{+}\right)$$
the time development equation for the order parameter ϕ is given by
(32)$$-\gamma \frac{\partial \phi }{\partial t}=-\kappa \frac{\partial^{2}\phi }{\partial x^{2}}+4g\phi \left(\phi -\phi_{-}\right)\left(\phi -\phi_{+}\right)$$
where
(33)$$\gamma =\frac{1}{\Gamma }$$

We assume the solution having the translational symmetry as
(34)$$\phi \left(x,t\right)=\psi \left(x-X\left(t\right)\right)$$
and substituting Equation (34) into Equation (32), we have
(35)$$\gamma \frac{dX}{dt}\psi^{\prime }=-\kappa \psi^{\prime \prime }+4g\psi \left(\psi -\phi_{-}\right)\left(\psi -\phi_{+}\right)$$

A kink-type solution of Equation (35) is given by [20,21]
(36)$$\psi \left(x-X\left(t\right)\right)=\frac{\phi_{+}}{1+e^{\alpha \phi_{+}\left(x-X\left(t\right)\right)}}$$
(37)$$\alpha =\sqrt{2g/\kappa }$$
where *X* must satisfy the equation
(38)$$\frac{dX}{dt}=V_{0}$$
with
(39)$$V_{0}=\frac{2g}{\gamma \alpha }\left(\phi_{+}-2\phi_{-}\right)=\frac{3g}{2\gamma \alpha }\left[\sqrt{1-\frac{8a_{G}}{g}}-1\right]$$

The kink-type solution given by Equation (36) behaves as $\psi \left(x-X\left(t\right)\right)\to \phi_{+}$ when x−X→−∞ and $\psi \left(x-X\left(t\right)\right)\to 0$ when x−X→+∞, and decreases sharply around x=X, as shown in Figure 3, because α is large (note that the value of κ is small). Therefore, x=X expresses the position of the sol–gel boundary, and the time development of the boundary is expressed by Equation (38). Equation (38) shows that the boundary motion is the uniform motion with velocity $V_{0}$. 

In the region $x>x_{f}$, the parameter a takes $a_{S}>0$ and then $\phi \left(x,t\right)=0$; the plasma is in the sol phase. Therefore, the gelation behavior is expressed by the motion of the boundary $x=X\left(t\right)$.

Although the inflow of gelators changes the value of the parameter a from $a_{S}$ to $a_{G}$, the gelation does not instantly start in the plasma. The gel nucleation growth at the interface between the gelator solution and plasma is required, and the finite lag time τ to complete the nucleation is also required. The lag time enables the forefront of the gel-forming polymer solution $x_{f}$ to precede the sol–gel boundary $X\left(t\right)$. Denoting the lag time for nucleation by τ, we obtain the initial condition of Equation (38) as
(40)$$X\left(\tau \right)=0$$

Therefore, the solution is summarized as
(41)$$\begin{array}{c} X=0\;\mathrm{when}\;0≦t≦\tau \\ X=V_{0}\left(t-\tau \right)\;\mathrm{when}\;\tau \;≦t \end{array}$$

Let us discuss the gel growth velocity $V_{0}$. From Equations (6) and (39), $V_{0}$ is given by
(42)$$V_{0}=\frac{2g}{\gamma \alpha }\frac{3\Delta f}{g\phi_{+}^{3}}=\frac{6\Delta f}{\gamma \alpha \phi_{+}^{3}}$$

In the above, the free-energy difference Δf given by
(43)$$\Delta f=f\left(0\right)-f\left(\phi_{+}\right)=-f\left(\phi_{+}\right)$$
expresses the stability of the gel state compared with the sol state of the gel-forming polymer solution. Equation (42) shows that the gel growth rate is larger as the gel state is more stable. 

### 2.4. Crossover of Dynamics

$X\left(t\right)$ is a linear function of t, while $x_{f}$ increases in proportion to $\sqrt{t}$. Therefore, the sol–gel boundary catches up with $x_{f}$ after a long elapsed time. After the catching-up time $t_{c}$, the gel-growth behavior is expressed by
(44)$$X\left(t\right)=x_{f}\left(t\right)$$

Hence, at time $t_{c}$, the gel growth behavior changes from the linear-time behavior $X\left(t\right)∼t$ to the square-root-time behavior $X\left(t\right)∼\sqrt{t}$. From Equations (30) and (41), the equation for time $t_{c}$ to satisfy is given by
(45)$$V_{0}\left(t_{c}-\tau \right)=\sqrt{2Kt_{c}}$$

The adequate solution of the equation is obtained as
(46)$$t_{c}=\tau +\frac{K}{V_{0}^{2}}-\sqrt{{\left(\frac{K}{V_{0}^{2}}\right)}^{2}+2\tau \left(\frac{K}{V_{0}^{2}}\right)}$$

When $t<t_{c}$, time is required for arrival of the sol–gel boundary for the gel-forming polymer solution to become gel. When $t>t_{c}$, the sol–gel boundary $X\left(t\right)$ and the gelator-diffusion front $x_{f}\left(t\right)$ are coincident and the solution becomes gel immediately after the precursor polymers are converted to gel-forming polymers by gelators. In general, the crossover time $t_{c}$ exists and the gel growth behavior changes from energy-limited growth to diffusion-limited growth as
(47)$$X\left(t\right)\;\sim \;t\;(\mathrm{energy}-\mathrm{limited:}\;t<t_{c})\;X\left(t\right)\;\sim \;\sqrt{t}\;(\mathrm{diffusion}-\mathrm{limited:}\;t>t_{c})$$

In cases where the gel nucleation time τ is very small, we have the behavior $X\left(t\right)\;\sim \;\sqrt{t}$ in the entire observed time range, and in cases where τ is very large, we have the behavior $X\left(t\right)\;\sim \;t$ in the entire observed time range, as shown in Figure 4.

Finally, let us pay attention to the coefficients of the two behaviors shown by Equation (47). The coefficient $V_{0}$ given by Equation (42) in the energy-limited behavior is expressed in terms of the quantities characterizing the activated plasma. It means that we can obtain information on the activated plasma by measuring the coefficient $V_{0}$. The coefficient K given by Equation (29) in the diffusion-limited behavior is expressed in terms of the quantities characterizing the gelator. Therefore, to obtain information on the gelator, we can analyze the coefficient K. 

## 3. Comparison of Experimental Results with Theory

Figure 5a shows photographs of the plasma gel layer grown from the tissue factor coating plate. When the plate was inserted in plasma, strongly turbid and harder gel layer of 0.3–0.6 mm thickness appeared immediately within 1 min and grew linearly proportional to time at the rate of around 0.05 mm/min for ca. 5 min. The thickness of the initial quick growth and the time for the linear growth were very sensitive to the operation. Then, a less turbid soft layer grew from it steadily to the end of the cell. Typical gray scale profiles of Figure 5a are shown in Figure 5b. The maximum of the gray scale decreased with time in the hard gel layer, while it was roughly constant in the soft gel layer. Strongly turbid layers were also observed for other systems using the tissue factor coating plate [22,23]. The thickness that can be compared with the length *X* studied in the theory developed in the preceding section should be that of the less turbid soft gel layer that was observed without uncertainty associated with the operation, i.e., the strongly turbid layer was regarded as the source of gelators in the theory. At the front of the soft gel layer, the gray scale decreased fairly sharply. The thickness *X* of the soft gel layer was determined as the distance between the two points determined from the intersection between the tangential line passing the point with the largest slope and the horizontal line (background), as shown in Figure 5c. 

Figure 6 shows the time course of *X* for plasma containing different [Ca^2+^]. No [Ca^2+^] dependence was observed in the experimental [Ca^2+^] range. The log–log plot consisted of two lines with different slopes: a slope around 1 at log *t* < 1.1 (*t* = 12 min) and one around 1/2 at log *t* > 1.1. The crossover behavior is consistent with the theoretical one shown in Equation (47). No [Ca^2+^] dependence of the curves indicated that [Ca^2+^] was not the parameter affecting the crossover time *t*_c_ in the experimental range of [Ca^2+^] near the physiological condition. *V*_0_ in Equation (41) and *K* in Equation (29) were estimated as 0.048 mm min^−1^ and 0.0250 mm^2^ min^−1^, respectively.

## 4. Conclusions

The aim of this project was to propose a method of analyzing gel growth behavior from a contact surface in plasma coagulation as an example of a system that consists of complex dynamics, including multiple reaction and diffusion processes. To reach the goal, we used a simplified model to describe the essence of blood coagulation. 

We considered the coagulation process with a model consisting of two processes; in the first process, the gelators inflow into plasma to activate it, while the second process is the gelation of the activated plasma. The plasma is thermodynamically expressed by the Ginzburg–Landau type free-energy functional in terms of the order parameter ϕ expressing the degree of gelation; the plasma is in the sol state when ϕ=0 and in the gel state when ϕ>0. The inflow gelators enable the gel to be in a thermodynamically stable state. Then, the value of the order parameter changes from ϕ=0 to $\phi =\phi_{+}>0$. 

The spatiotemporal behavior is described by the GL equation. The solution of the GL equation shows that there is a sharp boundary between the sol plasma and the gel plasma, and the gel growth behavior (and hence the blood coagulation behavior) can be expressed by the motion of the boundary. The gel width X(*t*) as a function of the elapsed time t is given by the position of the boundary.

The theory predicts the crossover behavior from the energy-limited gel growth $X\left(t\right)≃V_{0}t$ ($t<t_{c}$) governed by the thermodynamic property of the plasma to the diffusion-limited gel growth $X\left(t\right)≃\sqrt{2Kt}$ ($t>t_{c}$) governed by the diffusion of the gelator, where $t_{c}$ is the crossover time. We can obtain information on the activated plasma from the coefficient $V_{0}$ in the energy-limited growth and of the gelator from the coefficient *K*. 

A demonstration experiment was performed for plasma in contact with a tissue factor coating plate. The experiment showed the linear behavior X∼t in the early stage and the square root behavior $X∼\sqrt{t}$ in the late stage. Hence, the theoretical prediction was verified experimentally. The experimental verification for the proposed theory offers a novel method to analyze activated plasma and the gelator activating the plasma.

According to Kuprash et al. [23], the gel growth behavior correlates with deficiencies in blood coagulation proteins FV, FVII, FVIII, FIX, FX, and FXI, and some seem to be diffusion-limited while others are reaction-limited. It will be interesting to further analyze the different behaviors of the parameters *V*_0_ and *K* and the crossover time $t_{c}$. 

As the theory does not include special assumptions that are only applicable to blood coagulation, such as specific interactions between zymogens and serpins, and are instead implicitly renormalized in the parameters, the current crossover behavior obtained for plasma gelation could be applied to more general cases with complicated reaction- and diffusion-limited processes by focusing on the slowest reaction and diffusion processes.

## 5. Materials and Methods

For demonstration experiment, blood from healthy volunteers in 20 s was collected into 3.2% sodium citrate at a 9:1 *v*/*v* ratio. The anticoagulated blood was centrifuged at 3600 rpm (1600× *g*) for 10 min, and platelet-free plasma (PFP) was obtained from the supernatant. PFP was mixed with corn trypsin inhibitor (final concentration 0.2 mg/mL), and the desired amount of calcium chloride was mixed. 

Thrombodynamics kit (TDX-10, Hemacore, Moscow) containing an optical polymethylmethacrylate (PMMA) cell (1 mm × 7 mm × 15 mm), a tissue factor coating plate (100 pmol/m^2^), and corn trypsin inhibitor for minimizing contact activation was used. Then, 120 μL of plasma was poured into the optical cell, and the cell was settled in a water bath, the temperature of which was controlled at 37 °C. The gel growth experiment started by placing the tissue factor coating plate on top of the plasma. In the plasma, a turbid layer was grown from the contact surface with the tissue factor coating plate. The volume of the turbid layer was determined by an image analysis, and the gel volume was measured directly; these two volumes were found to be equivalent. Therefore, the growth of the gel layer was traced by observing the thickness of the turbid layer. 

## Figures and Tables

**Figure 1 gels-07-00011-f001:**
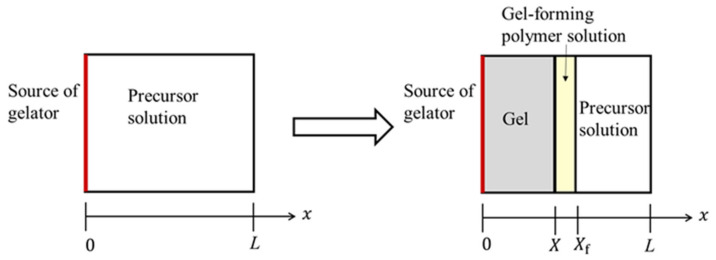
Illustration of the plasma (precursor solution) gel layer growth from the contact interface with the source of gelators. The *x*-axis is chosen to be perpendicular to the contact surface and is oriented in the direction from the contact interface to plasma. The origin of the *x*-axis is chosen at the contact interface. *X*(*t*) and *X*_f_(*t*) denote the thickness of the gel layer and gel-forming polymer solution, respectively, at the immersion time *t*.

**Figure 2 gels-07-00011-f002:**
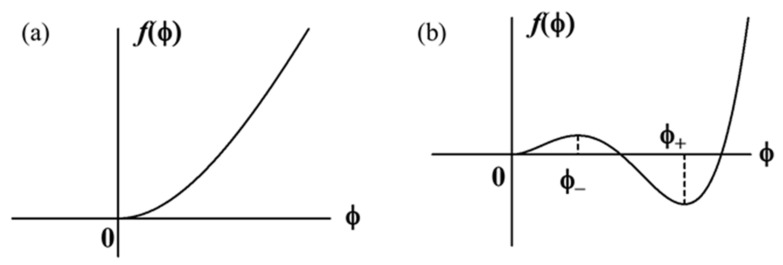
Local free energy of plasma for a > 0 (**a**) and a < 0 (**b**).

**Figure 3 gels-07-00011-f003:**
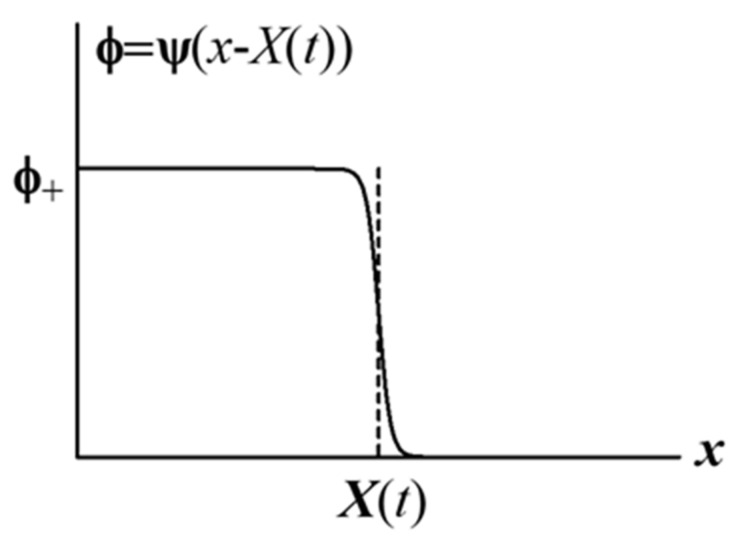
Kink-type spatial distribution of order parameter.

**Figure 4 gels-07-00011-f004:**
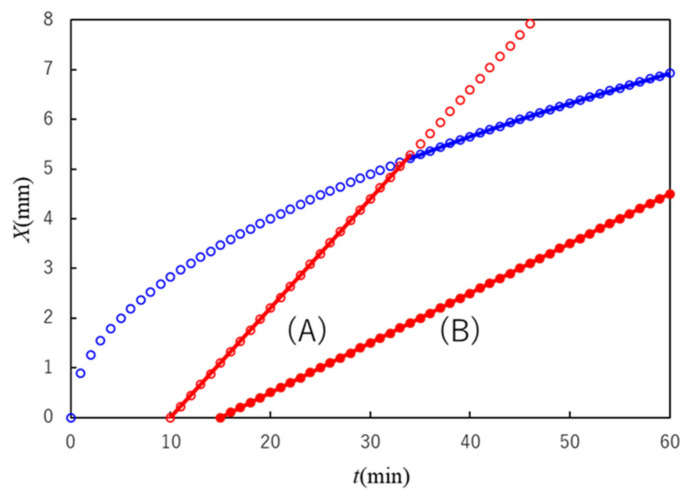
Illustration of the rate-limiting process in a multiple diffusion–reaction system in a limited experimental time range. The blue curve denotes the slowest diffusion-limited process in the system. When the slowest energy-limited process in the system is given by (A), the crossover from energy-limited to diffusion-limited process occurs at the intersection of the two curves. When the slowest energy-limited process is given by (B), the rate-limiting process is given by the line (B) in the whole process. The rate-limiting process is given by a solid curve (or line).

**Figure 5 gels-07-00011-f005:**
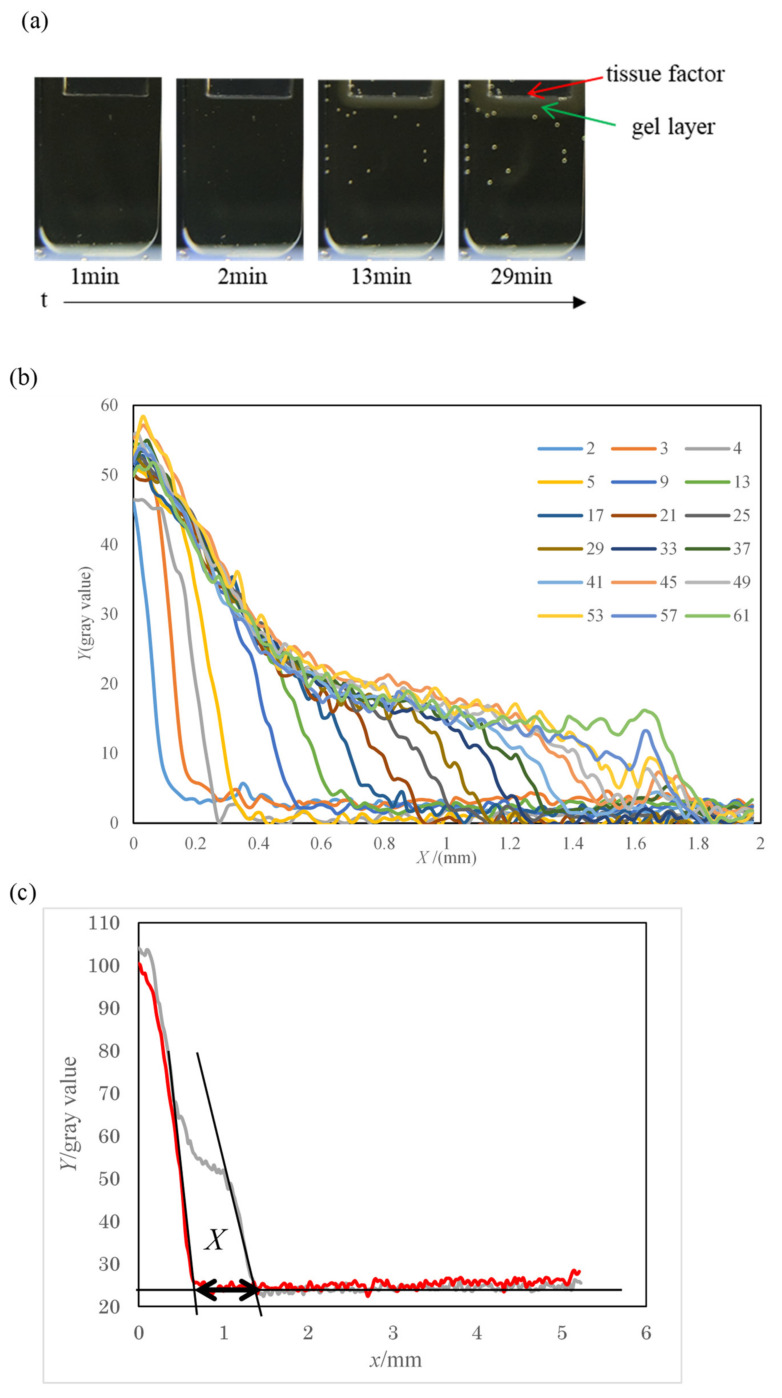
Photographs of gel grown from tissue factor coating plate (**a**), gray scale profiles (**b**), and definition of *X* as shown by the ↔ arrow (**c**) for [Ca^2+^] = 3 mM at indicated times.

**Figure 6 gels-07-00011-f006:**
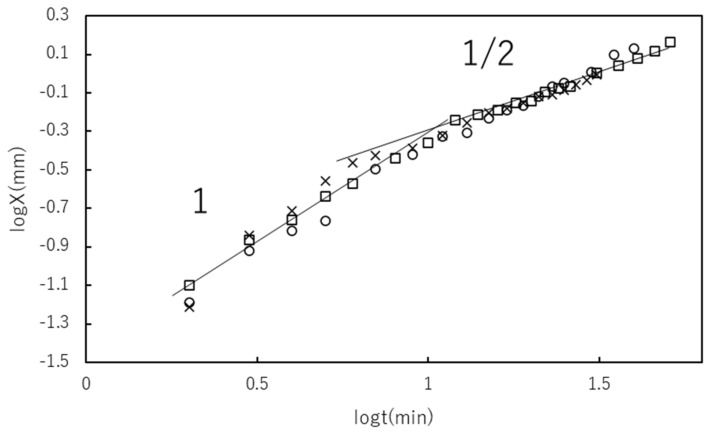
Time course of gel thickness X at various [Ca^2+^] = 3 (○), 7 (□) and 20 (×) mM in double logarithmic plot. Solid lines and slopes are given as a guide for the eyes.

## Data Availability

Data available on request due to restrictions, e.g., privacy or ethical.

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
