# Peer review of "Crossover of Rate-Limiting Process in Plasma Gel Growth by Contact with Source of Gelator"

_gels, 2021, doi:10.3390/gels7010011_

Round 1
Reviewer 1 Report
This paper is interesting, well argued and well written. The authors establish fairly well the investigation context for a problem that, despite being old, continues being of interest. The paper conclusion is not spectacular and to some extent is expected. However, I believe this paper is of sufficient quality to be published in GELS.
The model they propose is well founded. It is missed that the concepts and especially the parameters used in the development of the model are exposed with a greater physical meaning. Despite being rigorous, it is easy to lose the intuition while reading the paper. This is only a suggestion that I think, could improve the paper substantially.
A simple experiment corroborates the predictions. This is a good point.
The authors should remove Figure 6a. Figure 6b contains exactly the same information, and the power law exponents are clearly shown in the logarithmic version.
Finally, I do believe that the presence of COVID-19 in the article's conclusions is unnecessary, artificial and largely speculative. I think the paper is good enough, without the need of COVID-19.
Author Response
We removed Figure 6a and Covid-19 related sentences.
Reviewer 2 Report
This paper titled” Crossover of Rate Limiting Process in Plasma Gel Growth by Contact with Source of Gelator” is interesting, original and well written. The authors has developed a theory for the gel growth dynamics of plasma which is induced by contact with a source of gelators that are yielded due to the initiation, on the basis of the Ginzburg-Landau type dynamics with the gelator diffusion dynamics expressed by the moving boundary picture. The authors based on the theory which predicted the crossover of the rate limiting process in the time course of the thickness of the gel layer X from the energy-limited process expressed by ?∼? to the diffusion-limited process expressed by ?∼√?, from the contact of plasma with the source of gelators. Moreover the authors demonstrate and give nice experiment using putting a tissue factor coating plate as the initiator in plasma. The authors gives also the parameters characterizing plasma.
In my opinion, this paper is suitable for publication in Gels journal as it is, only some typographical errors should be corrected.
Author Response
We reviewed our manuscript thoroughly and corrected the typographical mistakes.